# Portrait Shadow Removal via Self-Exemplar Illumination Equalization

### Qian Huang*
South China University of Technology
serenohuang@gmail.com

### Cheng Xu*
South China University of Technology
cschengxu@gmail.com

### Guiqing Li
South China University of Technology
ligq@scut.edu.cn

### Ziheng Wu
Platform of AI (PAI), Alibaba Group
ziheng.wzh@alibaba-inc.com

### Shengxin Liu
Harbin Institute of Technology,
Shenzhen
sxliu@hit.edu.cn

### Shengfeng He[†]
Singapore Management University
shengfenghe@smu.edu.sg

## Abstract

We introduce the Self-Exemplar Illumination Equalization Network, designed specifically for effective portrait shadow removal. The core idea of our method is that partially shadowed portraits can find ideal exemplars within their non-shadowed facial regions. Rather than directly fusing two distinct classes of facial features, our approach utilizes non-shadowed regions as an illumination indicator to equalize the shadowed regions, generating deshadowed results without boundary-merging artifacts. Our network comprises cascaded Self-Exemplar Illumination Equalization Blocks (SExmBlock), each containing two modules: a self-exemplar feature matching module and a feature-level illumination rectification module. The former identifies and applies internal illumination exemplars to shadowed areas, producing illumination-corrected features, while the latter adjusts shadow illumination by reapplying the illumination factors from these features to the input face. Applying this series of SExmBlocks to shadowed portraits incrementally eliminates shadows and preserves clear, accurate facial details. The effectiveness of our method is demonstrated through evaluations on two public shadow portrait datasets, where it surpasses existing state-of-the-art methods in both qualitative and quantitative assessments.

## CCS Concepts

• **Computing methodologies → Computational photography**; **Image processing**.

## Keywords

Portrait Shadow Removal, Self-Exemplar Illumination Equalization, Correspondence Feature Matching

---

*Both authors contributed equally to the paper.
†Corresponding author.

---

**ACM Reference Format:**
Qian Huang, Cheng Xu, Guiqing Li, Ziheng Wu, Shengxin Liu, and Shengfeng He. 2024. Portrait Shadow Removal via Self-Exemplar Illumination Equalization. In *Proceedings of the 32nd ACM International Conference on Multimedia (MM '24), October 28-November 1, 2024, Melbourne, VIC, Australia.* ACM, New York, NY, USA, 9 pages. https://doi.org/10.1145/3664647.3681000

## 1 Introduction

Portrait photography, a genre focused on capturing an individual's expressions, appearances, and unique characteristics, is both an art form and commercially valuable. In today's digital age, where the internet and social media are integral to daily life, portrait photography has become increasingly important. However, the challenging illumination conditions in real-world settings, influenced by both natural and artificial light sources, often result in portrait images with severe shadow occlusions. These shadows not only diminish the visual quality and artistic value of the photographs but also present significant obstacles for various facial-related tasks such as face recognition and expression estimation.

Over the past decades, numerous methods have been proposed to remove shadows from portrait images. Traditional techniques primarily rely on histogram manipulation [8, 13, 26], color transfer [20, 36, 43, 49], or illumination modeling/compensation [1, 9, 38]. While effective for simple shadow patterns, their performance significantly declines with more complex shadows. Addressing this, recent research has shifted towards deep learning-based solutions for higher quality shadow removal [15, 27–29, 51]. For instance, Hu *et al.* [15] leverage direction-aware spatial context for shadow detection and removal. Zhang *et al.* [51] introduces two models to address foreign and facial shadows. Liu *et al.* [29] decompose the RGB shadow removal problem into grayscale shadow removal and colorization.

Notwithstanding the demonstrated success, existing shadow removal methods still struggle with artifacts near shadow/non-shadow boundaries and facial distortions, presenting significant challenges. Most current techniques [28, 30, 53] focus on learning a transformation from the shadow domain to a shadow-free domain. However, the diverse shadow conditions and facial complexions in portrait images make it difficult for these methods to learn an accurate and consistent mapping. Consequently, they often result in non-uniform illumination and obscured facial details in

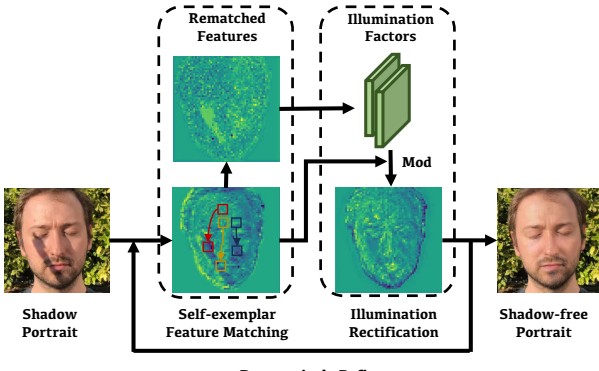

**Figure 1: We present an innovative approach for portrait shadow removal, designed to achieve precise shadow elimination while meticulously preserving facial details. Our method fully utilizes internal illumination exemplars, progressively balancing the illumination differences between shadowed and non-shadowed regions.**

their outputs. To avoid the requirement of learning from extensive datasets, He *et al.* [12] suggest using the generative priors in a pretrained StyleGAN [19] to restore uniform portrait illumination. While innovative, this approach is computationally intensive and time-consuming, as it necessitates optimization for each input separately. Furthermore, it frequently leads to severe facial distortions due to inadequate latent code optimization. In pursuit of improved portrait shadow removal, we note that portraits captured in real-world settings typically exhibit only partial shadow occlusions. Consequently, the non-shadowed facial regions can offer substantial illumination priors. These priors can act as valuable guidance, enabling more accurate shadow removal while ensuring the preservation of facial details.

Motivated by the above observation, we introduce the Self-Exemplar Illumination Equalization Network (SExmNet). This network employs a series of Self-Exemplar Illumination Equalization Blocks (SExmBlocks) to progressively eliminate shadow occlusions. Each block features a bespoke self-exemplar feature matching mechanism, designed to estimate a shadow-oriented matching correspondence map from the input features. Utilizing this map, illumination information from the non-shadow region is precisely transferred to shadowed areas, leading to illumination-rectified, rematched features.

However, these rematched results often display non-smooth facial patterns due to the warping process. Instead of directly using these features to generate the output, we distill illumination information from the rematched features into two spatial matrices of illumination factors. Reapplying these factors to the input facial features allows for the gradual removal of shadow occlusions, ensuring the retention of realistic facial details. Our extensive experiments across two public datasets confirm SExmNet's superiority over current state-of-the-art methods, highlighting its effectiveness in both shadow removal and detail preservation.

In summary, our main contributions are three-fold:

- We introduce SExmNet, an innovative approach leveraging internal facial illumination cues for portrait shadow removal. To our knowledge, this is the first initiative to address portrait shadow removal using internal feature matching, achieving realistic shadow elimination while preserving facial details.
- We tailor the SExmBlock, specifically for shadow-oriented correspondence matching and facial illumination rectification. This block facilitates precise shadow removal and the recovery of authentic facial textures.
- Extensive experiments confirm that our method outperforms state-of-the-arts, demonstrating its superiority in both shadow removal and detail preservation.

## 2 Related Work

**Shadow Removal.** Early studies in shadow removal primarily investigated shadows' physical properties. These methods began by detecting shadows using illumination discontinuity and color inconsistency at shadow edges [2], followed by shadow removal through histogram manipulation [8, 13, 26], style transfer [20, 36, 43, 49], or illumination compensation [1, 9, 38]. However, their efficacy is limited to simple shadows and they falter in complex scenes. Recently, with the significant advancement of deep learning in image synthesis and restoration [44–46], various deep learning-based methods have been also introduced to rectify shadow illumination more convincingly [5, 7, 15, 16, 24, 27, 28, 35, 41]. While these approaches yield visually superior results compared to classical ones, they struggle with non-uniform illumination and loss of facial details, as directly learning a precise and consistent mapping from a complex illumination and facial pattern distribution is challenging. Our proposal overcomes these issues by utilizing internal illumination exemplars for guided shadow removal, simplifying the learning process and enhancing result quality.

**Illumination Compensation.** This process aims to improve object visibility in images by addressing uneven lighting. Traditional techniques often involve histogram manipulation [8, 25] and Retinex theory [22, 23], generally manipulating illumination globally or requiring meticulous parameter tuning, leading to significant detail loss and distortions. In recent years, deep learning methods [3, 4, 11, 14, 25, 39] have been developed for more compelling illumination correction. Despite their advancements, they often compromise facial details. In contrast, our method accurately rectifies local illumination while preserving original facial details.

**Matching Correspondence** refers to the pixel or image patch correspondence between images. Early works like HOG [6] and SIFT [31] estimate this correspondence at the image or feature level. Recently, deep learning has led to more robust correspondence prediction, facilitating various tasks like image translation [50], super-resolution [32], and view synthesis [52]. In face synthesis, researchers have explored matching correspondence to improve synthesis quality. Xu *et al.* [47] address large-angle face synthesis by breaking it down into smaller-angle rotations, combining GAN with face flow methods, while Wei *et al.* [42] propose a Flow-based Feature Warping Model for synthesizing realistic, illumination-consistent human face images. Unlike these studies, we make the

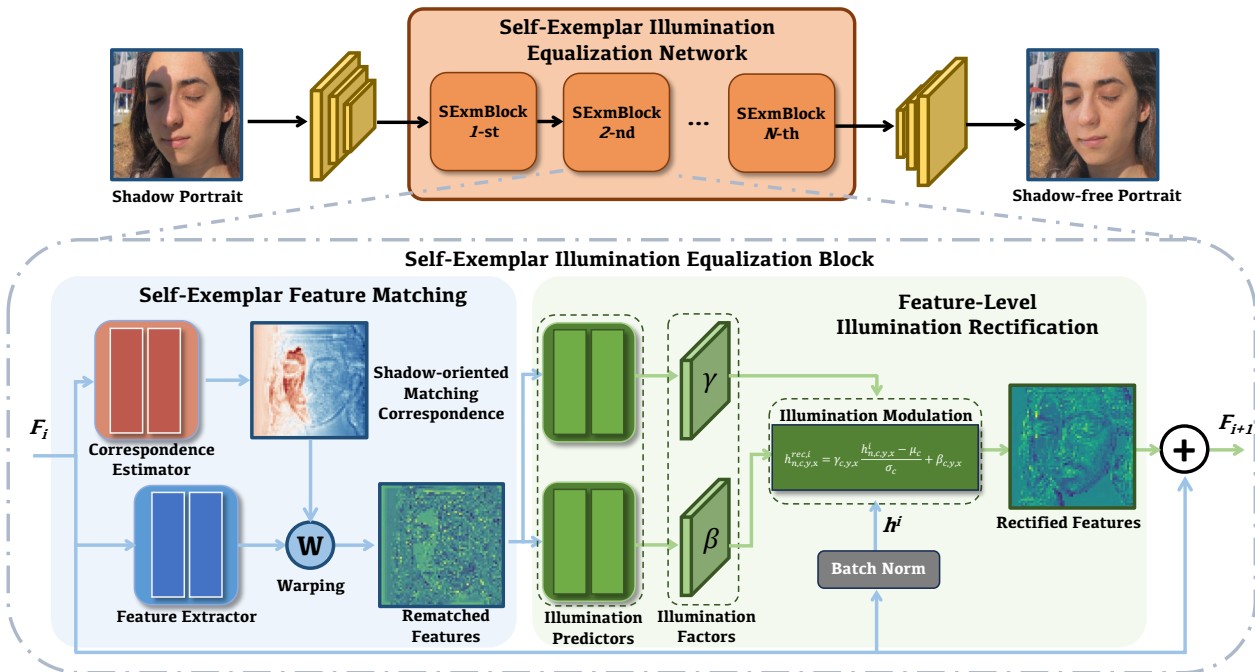

**Figure 2: Overview of the generator in our method. The Self-Exemplar Illumination Equalization Network (SExmNet) inputs a shadowed portrait image and incrementally eliminates shadow occlusions. It achieves this by sequentially applying self-exemplar illumination equalization blocks (SExmBlock) to the facial features in the image. Each block harnesses internal illumination cues, performing both self-exemplar feature matching and illumination rectification to correct shadowed areas. The output features from the final block are used to produce the end result, effectively removing shadows from the portrait.**

first attempt to explore unsupervised matching correspondence in portrait shadow removal, significantly elevating removal quality.

## 3 Method

### 3.1 Formulation and Overview

Our goal is to fully excavate and exploit internal illumination exemplars as explicit guidance to achieve precise portrait shadow removal while restoring realistic and faithful facial details. To this end, we have two domains with paired data, *i.e.*, $X_s = \left\{x_s^i \mid x_s^i \in X_s\right\}_{n=i,...,N}$ and $X_f = \left\{x_f^i \mid x_f^i \in X_f\right\}_{n=i,...,N}$, representing the shadow and the shadow-free portrait domains, respectively. Here, $\left(x_s^i, x_f^i\right)$ constitutes a pair for training. Given an input $x_s$, we aim to learn a mapping function from the shadow portrait domain to the shadow-free portrait domain: $x_f = \mathcal{G}(x_s)$, where $x_f$ is the shadow removal result of $x_s$.

The framework of our model is illustrated in Fig. 2. Taking $x_s$ as input, our model first extracts the facial features $F_0$ from $x_s$. Then $F_0$ is fed into the Self-Exemplar Illumination Equalization Network (SExmNet) , which contains a cascade of Self-Exemplar Illumination Equalization Blocks (SExmBlock). Here, each SExmBlock comprises a self-exemplar illumination feature matching module and a feature-level illumination rectification module, which are responsible for explicitly leveraging the internal illumination exemplars for relighting the shadow region, and rectifying the illumination of the

input facial features, respectively. By progressively removing the shadow occlusions via the cascaded SExmBlocks, the output of the last SExmBlock is finally fed into a decoder to render a visually plausible shadow-free portrait image $\hat{x}_f$. We discuss the details of each component in the following sections.

### 3.2 Self-Exemplar Feature Matching

To fully exploit the illumination cues from non-shadow regions to facilitate precise shadow removal, we propose a Self-Exemplar Feature Matching module (SExmFM) for identifying and applying internal illumination exemplars to the shadowed regions. The SExmFM module is designed to learn a shadow-oriented matching correspondence map between the shadowed and non-shadowed regions. This map can be used to guide the illumination propagation from the non-shadowed regions to the shadowed regions. Specifically, assuming $i$ denotes the index of the SExmBlock, given the facial features $F_i$ as input, a correspondence estimator is first employed to estimate the shadow-oriented matching correspondence map $M_i$ ($M_i \in \mathbb{R}^{H \times W \times 2}$), which can be formulated as

$$M_i = E_c(F_i), \tag{1}$$

where $E_c$ represents the correspondence estimator. The two channels in each map $M_i$ correspond to offsets in the $x$ and $y$ directions for feature matching.

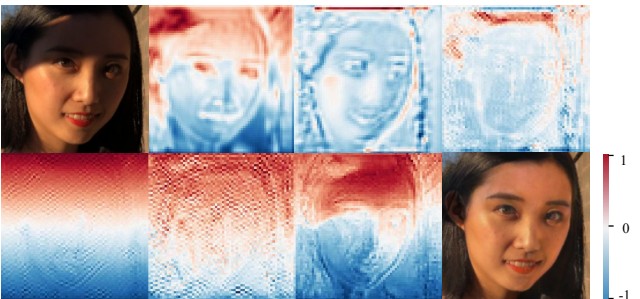

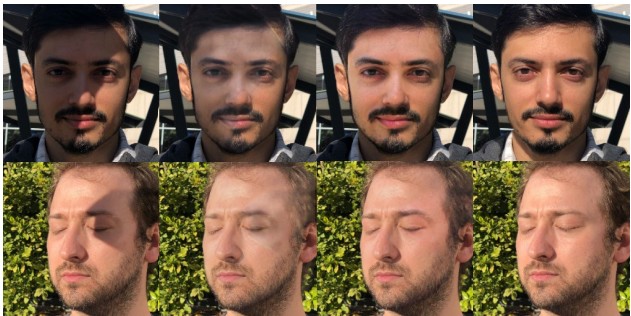

| Input | w/o Illumination Rectification | Full model | Ground truth |

**Figure 4: Visualization of the effects of the feature-level illumination rectification. Omitting illumination rectification can lead to significant facial distortions and artifacts.**

**Figure 3: Visualization of shadow-oriented matching correspondence maps. We showcase six learned shadow-oriented matching correspondence maps from corresponding six self-exemplar illumination equalization blocks (from left to right and top to bottom). The input and output portraits are also shown for reference. Our model can indeed remove the shadow occlusions by progressively propagating the illumination information from the non-shadow regions to the shadow regions. Note that the number of the self-exemplar illumination equalization blocks is a fixed hyper-parameter.**

Then, this correspondence map $M_i$ is used to warp the original facial feature $F_i$ to the rematched features $F_{rem}^i$. This process can be presented as follows:

$$F_{rem}^i = \mathcal{W}(F_i, M_i), \tag{2}$$

where $\mathcal{W}$ denotes the differentiable warping operation, which is a grid-sample operator in this work.

Note that, the correspondence map $M_i$ can be simply learned with indirect supervision from the final shadow-removed image. The rationale behind this is that the shadowed and non-shadowed regions in the input portrait image are highly correlated, and this can offer strong guidance for the correspondence map learning. Fig. 3 depicts the visualization of the learned shadow-oriented matching correspondence map in each SExmBlock. Here, each pixel in the correspondence map indicates the matching offset between the current pixel and its reference pixel. We can observe that the correspondence map indeed shows accurate matching relationships between the shadow and non-shadow regions, which serves as strong guidance for illumination propagation from the non-shadow regions to the shadowed regions. Particularly, the upper and right part of the input portrait is occluded by the shadows. As a consequence, we can see that the shadowed pixels in the correspondence map are mostly matched to the non-shadowed pixels in the lower and left part of the face. Under this explicit guidance, the illumination information from the lower and left part can be effectively propagated to the shadowed region in the upper and right part of the face, rendering the final illumination-rectified result.

### 3.3 Feature-Level Illumination Rectification

Although self-exemplar feature matching allows our model to explicitly leverage the internal illumination priors to compensate the shadow occlusions, it also introduces undesired facial feature distortions and artifacts to the rematched features due to the explicit

warping operation. Therefore, it is infeasible to directly employ the rematched features to generate the final result. As presented in Fig. 4, when we remove the illumination rectification module and directly use the rematched features to render a shadow-free portrait image (w/o Illumination Rectification), the generated result suffers from facial distortions and unnatural illumination effects.

To fully exploit the rectified illumination information from the rematched features while avoiding introducing unnatural facial features, we propose a feature-level illumination rectification module (FLIR) to explicitly distill the illumination information from the rematched features, and then adapt it to the input facial features. As shown in the green part of Fig. 2, the illumination rectification process consists of two steps, i.e., the illumination factors prediction and the illumination modulation. The former aims to grasp the illumination information from the rematched features, which can be subsequently utilized by the latter step for modulating the input facial features to recover equalized illumination.

Specifically, the rematched features $F_{rem}^i$ are first fed into two different convolution layes for predicting the illumination factors $\gamma$ and $\beta$, respectively. Here, $\gamma$ and $\beta$ are two spatial matrices with the same size as $F_{rem}^i$, which can be used as modulation parameters to adapt the illumination information to the input facial features via a spatially-adaptive denormalization operation [34]. In this way, the non-uniform illuminations can be effectively mitigated without sacrificing the the original structure and details of the face features. This is the key to producing compelling shadow-removed results with well-preserved faithful and rich facial details. Formally, let $h^i$ denote the activations of the current normalization layer for a batch of $N$ samples. $C$, $H$, and $W$ denote the number of channels, the height, and the width of the activation maps in the layer. The illumination modulation process can be calculated as follows:

$$h_{n,c,y,x}^{rec,i} = \gamma_{c,y,x} \frac{h_{n,c,y,x}^i - \mu_c}{\sigma_c} + \beta_{c,y,x}, \tag{3}$$

where $h_{n,c,y,x}^i$ and $h_{n,c,y,x}^{rec,i}$ denote the activation value before and after modulation, respectively. $\mu_c$ and $\sigma_c$ denote the mean and standard deviation of the activations in channel $c$.

Once the illumination modulation is performed, we can get the illumination-rectified features $F_{rec}^i$, which is fused with the original $F_i$ and then sent to the next SExmBlock for further illumination equalization. The fusion process can be formulated as follows:

$$F_{i+1} = F_i \oplus F_{rec}^i, \tag{4}$$

where $\oplus$ denotes the element-wise addition operation.

## 3.4 Loss Functions

To produce photo-realistic shadow-free portraits with well-preserved faithful facial details, we adopt three simple yet effective losses, including the pixel-wise loss, perceptual loss, and the adversarial loss to govern the training in an end-to-end manner. All the losses are directly applied on the generated shadow-free portrait image $\hat{x}_f$.

**Pixel-wise Loss.** To maintain the content consistency, we apply a pixel-wise loss by minimizing the $L1$-distance between the generated shadow-free portrait image $\hat{x}_f$ and the ground truth $x_f$:

$$\mathcal{L}_{pix} = \| \hat{x}_f - x_f \|_1 . \tag{5}$$

**Perceptual Loss.** We also introduce a perceptual loss [17] to improve visual quality of the generated results, it computes the $L1$-distance between extracted features of the generated shadow portrait image $\hat{x}_f$ and the ground truth $x_f$:

$$\mathcal{L}_{per} = \sum_i \| \Phi_i(\hat{x}_f) - \Phi_i(x_f) \|_1, \tag{6}$$

where $\Phi(\cdot)$ denotes the extracted features from certain layers in the VGG-19 model [40], which is pre-trained on the ImageNet classification dataset [37] and has a powerful feature extraction capability. Following [33], we choose the first four layers of VGG-19 for computing the perceptual loss.

**Adversarial Loss.** We adopt a ResNet-based discriminator $D$ to provide adversarial supervision signals, such that the generator can produce more photo-realistic results. The adversarial loss is formulated as follows:

$$\mathcal{L}_{adv} = \mathbb{E}\left[ \log D\left(x_f\right) + \log \left[1 - D\left(\hat{x}_f\right)\right] \right] . \tag{7}$$

**Total Loss.** The total loss for training our model is a combination of the pixel-wise loss, perceptual loss, and adversarial loss:

$$\mathcal{L}_{total} = \lambda_1 \mathcal{L}_{pixel} + \lambda_2 \mathcal{L}_{per} + \lambda_3 \mathcal{L}_{adv}, \tag{8}$$

where $\lambda_1$, $\lambda_2$, and $\lambda_3$ are weighting parameters to balance the corresponding loss items.

## 4 Experiments

### 4.1 Settings

**Implementation Details.** Our model is implemented on the PyTorch framework on one NVIDIA A100 GPU with 80 GB of memory. We adopt the Adam optimizer [21] ($\beta_1 = 0.5$, $\beta_2 = 0.999$) with the initial learning rate set as 0.0002 for our training, which is linearly decayed as the training proceeds. The training batch size is set to 16. For both the training and testing, the images are all resize to $256 \times 256$. We empirically set $\lambda_1$, $\lambda_2$, and $\lambda_3$ as 5, 5, and 1, respectively. The number of SExmBlock is set as 6. To enable more

**Table 1: Quantitative comparison with the state-of-the-art methods on the UCB [51] dataset. We achieve the best results among all the competitors.**

| Method | SSIM ↑ | PSNR ↑ | LPIPS ↓ |
|---|---|---|---|
| Guo *et al.* [10] | 0.605 | 14.205 | 0.279 |
| He *et al.* [12] | 0.732 | 20.001 | 0.110 |
| Hu *et al.* [15] | 0.774 | 20.532 | 0.097 |
| Cun *et al.* [5] | 0.784 | 21.307 | 0.093 |
| Zhang *et al.* [51] | 0.782 | 23.816 | 0.074 |
| Liu *et al.* [29] | 0.851 | 23.738 | 0.068 |
| DMTN [27] | 0.783 | 21.574 | 0.085 |
| TBRNet [28] | 0.790 | 22.299 | 0.082 |
| Ours | **0.883** | **24.174** | **0.059** |

accurate matching correspondence learning, all the backgrounds of the input face features are masked out by using a facial mask predicted by a pretrained face parsing model [48]. The feature extractor consists of a convolutional layer with 128 output channels. The correspondence estimator comprises three convolutional layers, each using a $3 \times 3$ kernel, with a stride of $1 \times 1$ and padding of $1 \times 1$. The numbers of output channels are [64, 32, 2], respectively. Between each convolutional layer, there is a LeakyReLU layer.

**Datasets.** Since there is no publicly available large-scale paired shadow/shadow-free dataset, we follow [29] to synthesize our training data from CelebA-HQ [18], by combining shadow-free portraits with pre-defined shadows masks.

Specifically, we synthesize 3,000 pairs of shadow/shadow-free portrait images for our training. To evaluate our method, we follow the same protocal as [51], which performs both qualitative and quantitative evaluations on UCB [51], a dataset consisting of a very limited number of 100 paired shadow/shadow-free face images. To further verify the generalization capabiltity of our method, we also conduct evaluations on the Shadow Faces in the Wild dataset (SFW [29]), which includes 280 videos from 20 subjects.

**Metrics.** For quantitative evaluations, we choose three widely used metrics, including structure similarity measure (SSIM), peak signal-to-noise ratio (PSNR), and Learned Perceptual Image Patch Similarity (LPIPS). In particular, SSIM and PSNR are used to evaluate overall quality of portrait shadow removal, while LPIPS measures the perceptual quality of the shadow-removed and the ground truth shadow-free portrait images. Moreover, we also involve the video-based Frechet inception distance (VFID) and flow warping error ($E_{warp}$) to measure the perceptual quality and temporal coherence of shadow-removed videos, respectively.

### 4.2 Comparison with State-of-the-arts

For a comprehensive comparison, we compare our framework to eight state-of-the-art methods, including five general shadow removal methods [5, 10, 15, 27, 28], and three portrait shadow removal methods [12, 29, 51] both qualitatively and quantitatively.

**Qualitative Comparison.** We first compare our method with eight competitors quantitatively on the UCB [51] dataset. Fig. 5 showcases the visualization results of shadow removal from different methods. As can be observed, all the compared methods

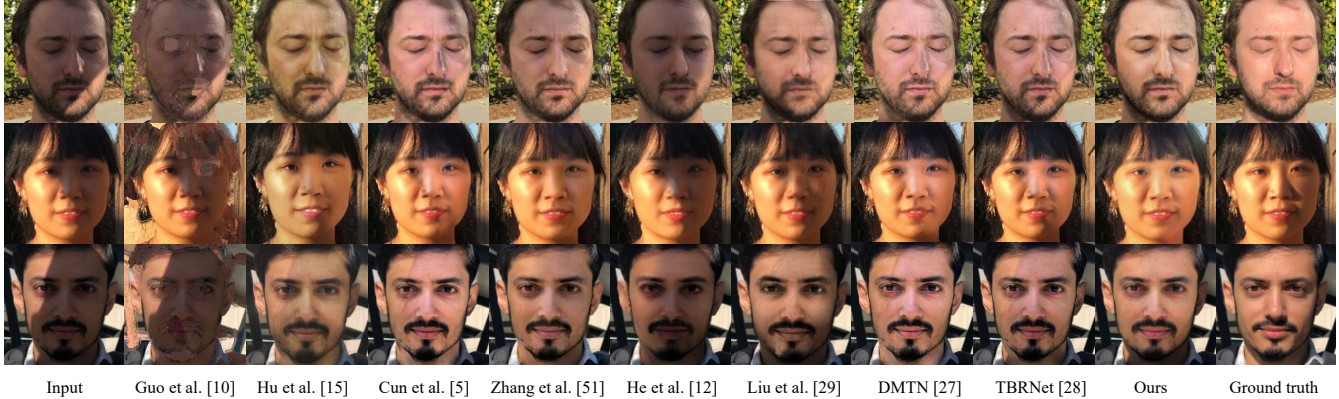

Input   Guo et al. [10]   Hu et al. [15]   Cun et al. [5]   Zhang et al. [51]   He et al. [12]   Liu et al. [29]   DMTN [27]   TBRNet [28]   Ours   Ground truth

**Figure 5: Qualitative comparison with the state-of-the-art methods on the UCB [51] dataset. Our method best preserves facial details while seamlessly removing facial shadows. Note that the ground truth shadow-free images of some UCB samples are not perfect as them originally contain slight shadows.**

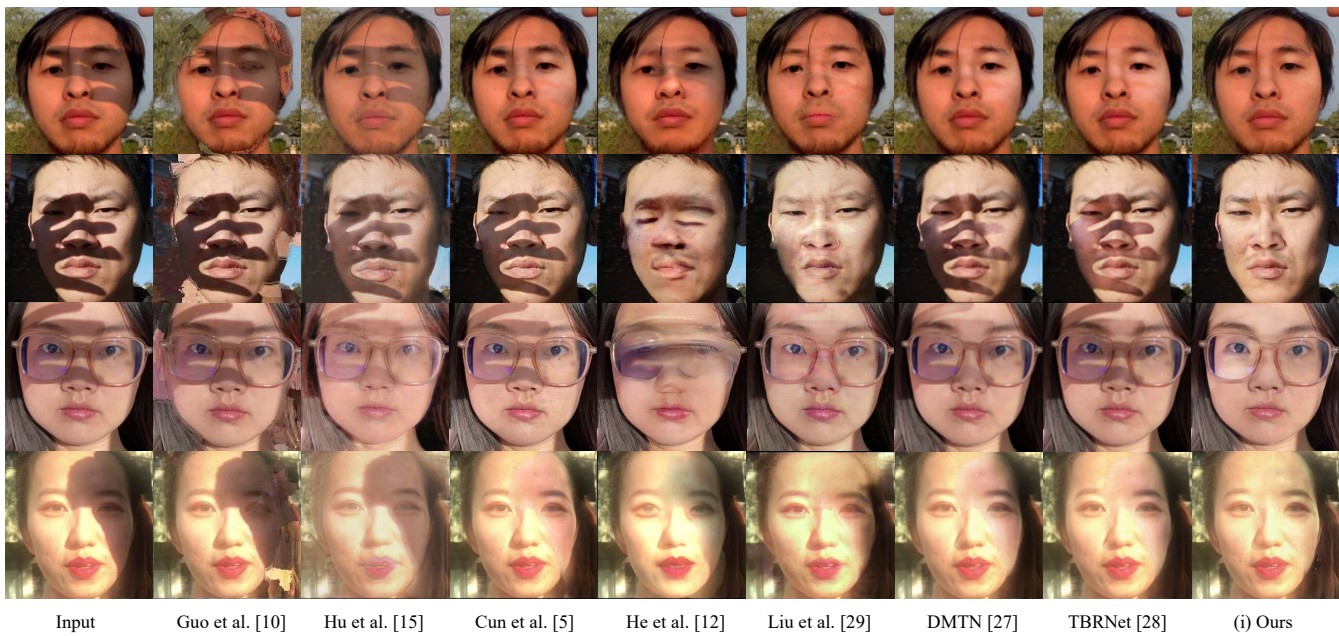

Input   Guo et al. [10]   Hu et al. [15]   Cun et al. [5]   He et al. [12]   Liu et al. [29]   DMTN [27]   TBRNet [28]   (i) Ours

**Figure 6: Qualitative comparison with the state-of-the-art methods on the SFW [29] dataset. Note that the ground truth shadow-free images are not available for this dataset.**

struggle to produce visually plausible shadow-removed results. In particular, the traditional method Guo *et al.* [10] produces severe facial pattern artifacts and distortions as it is built upon a simple parameterized shadow removal model, which is not able to deal with complex shadows in real-world scenarios. Hu *et al.* [15] struggles to rectify the shadow illuminations. Although Guo *et al.* [10] succeeds in relighting the shadow regions, the resulting illuminations are significantly inconsistent with the non-shadow regions. He *et al.* [12] is unable to remove the shadows and tend to incur undesired changes of facial features, this is because it completely

rely on generative priors for shadow removal by performing latent code optimization, leading to poor generalization capability on real-world shadows and inevitable contamination on the original facial features. Cun *et al.* [5], DMTN [27], and TBRNet [28] struggle with non-uniform illuminations. Although Liu *et al.* [29], and Zhang *et al.* [51] can deliver relatively naturally-looking results, they still undergo noticeable shadow residues. Different from all the above competitors, our method can create visual-pleasing shadow removal results with uniform illuminations and faithful facial details. We mainly attribute this to the delicate exploitation of internal illumination cues and decent illumination rectification

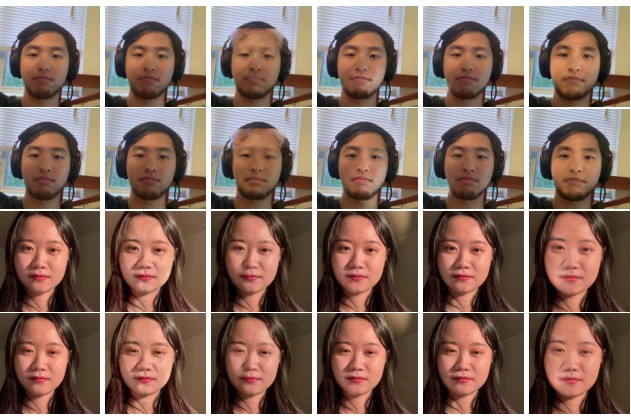

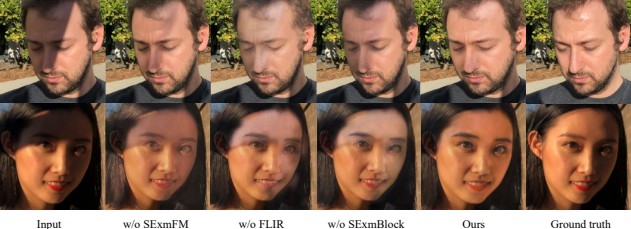

Input    w/o SExmFM    w/o FLIR    w/o SExmBlock    Ours    Ground truth

**Figure 8: Qualitative comparison of different variants of our method on the UCB [51] dataset. Each of our component contributes to our final model.**

Input    Cun et al. [5]    Liu et al. [29]    DMTN [27]    TBRNet [28]    Ours

**Figure 7: Examples of portrait video shadow removal on the SFW [29] dataset. We show two cases and each case includes two inconsecutive frames as input and their corresponding shadow-removed results of different methods. Our method delivers higher-quality shadow-free results with better temporal consistency than the other approaches. Due to limited space, only the results of the top four baselines in terms of quantitative performance are shown here. Please refer to our supplementary videos for complete video comparison with all compared methods.**

mechanism, which effectively loosen the learning difficulty and facilitate the shadow removal process. Moreover, it is also worth noting that our method can perfectly remove both the external or internal shadows with consistent illuminations and vivid facial details, which is not achieved by any of the competitors.

To further demonstrate the generalization capability of our method, we also conduct qualitative evaluations on the SFW [29] dataset. As shown in Fig. 6, all the compared methods struggle to create satisfactory shadow removal results given shadow portraits captured in highly dynamic and complex real-world scenes, while our method can consistently produce compelling shadow-free results. This verifies the superior generalization capability of our model and implies its huge potential for real-world applications.

**Quantitative Comparison.** We also perform a quantitative comparison with the existing methods. As shown in Table 1, our method provides noticeable improvements on all of the three quantitative metics. This indicates that our method can indeed effectively recover the illuminations and facial details, yielding better results in terms of both image and perceptual quality. This is consistent with our visualization results.

### 4.3 Portrait Shadow Removal on Videos

Fig. 7 presents the shadow removal comparison results of different methods on portrait videos. Benefiting from the effective utilization of internal illumination exemplars, our method can produce visually plausible shadow removal results with consistent illuminations and faithful facial details across video frames, compared to other competitors. The quantitative performance of different methods on video shadow removal are also provided in Table 2,

which reveals that our model can indeed render visually convincing shadow-free videos, leading to the best VFID value among all the compared methods. Please refer to our supplementary video for better assessment.

### 4.4 Ablation Study

In this section, we conduct an in-depth ablation study to validate the efficacy of our main proposals, including self-exemplar feature matching, feature-level illumination rectification, and self-exemplar illumination equalization block.

**Self-Exemplar Feature Matching.** To verify the effectiveness of Self-Exemplar Feature Matching (SExmFM), we remove the self-exemplar feature matching module by directly feeding the input features to the illumination rectification module. As shown in the second column of Fig. 8, the result suffers from noticeable shadow residues as this model variant falls short of exploiting the internal illumination exemplars, and thus struggles to recover equalized illuminations in the shadowed region. Table 3 also shows that the model variant without self-exemplar feature matching performs worse than our full model in terms of all three quantitative metrics.

**Feature-Level illumination Rectification.** We also investigate the efficacy of feature-level illumination rectification (FLIR) by omitting the illumination rectification module. As can be seen in the third column of Fig. 8, severe non-uniform illuminations artifacts and facial distortions exist in the result. The reason behind this is that the warping operation applied in self-exemplar feature matching inevitably introduces notable facial distortions and artifacts, and simply integrating the rematched features into the input facial features can result in unnatural illuminations and contaminations on the original facial details. In contrast, our feature-level illumination rectification module can distill the recovered illumination information and compensate the shadow illuminations in the input facial features. This not only allows for precise shadow removal, but also well retain the original facial details, which is also demonstrated quantitatively in Table 3.

**Self-Exemplar Illumination Equalization Block.** To get a deeper insight into the significance of the Self-Exemplar illumination Equalization Block (SExmBlock), we replace all the SExmBlocks with standard residual blocks in our network. As observed in the fourth column of Fig. 8, the result undergoes unnatural illuminations and facial artifacts. This is because the standard residual blocks are not able to explicitly leverage the internal illumination

**Table 2: Quantitative comparison with the state-of-the-art methods on the SFW [29] dataset. We achieve the best results among all the competitors.**

| Method | VFID ↓ | $E_{warp}$ ↓ |
|---|---|---|
| Guo *et al.* [10] | 0.930 | 0.0111 |
| He *et al.* [12] | 0.370 | 0.0075 |
| Hu *et al.* [15] | 0.398 | 0.0049 |
| Cun *et al.* [5] | 0.353 | 0.0043 |
| Liu *et al.* [29] | 0.300 | 0.0047 |
| DMTN [27] | 0.324 | 0.0045 |
| TBRNet [28] | 0.335 | 0.0047 |
| Ours | **0.289** | **0.0040** |

**Table 3: Quantitative comparison among different variants of our method on the UCB [51] dataset. Each of our component contributes to our final model.**

| Method | SSIM ↑ | PSNR ↑ | LPIPS ↓ |
|---|---|---|---|
| w/o SExmFM | 0.866 | 23.383 | 0.068 |
| w/o FLIR | 0.858 | 23.251 | 0.079 |
| w/o SExmBlock | 0.840 | 22.708 | 0.075 |
| Full model (Ours) | **0.883** | **24.174** | **0.059** |

priors and therefore prone to learning an unstable and inconsistent mapping, leading to inferior shadow removal results (Table 3). By fully exploiting the internal illumination exemplars and accurately propagate the illuminations to the shadow regions, our SExmBlocks enables the model to learn a more accurate and consistent mapping that delivers visually plausible shadow removal results with uniform illuminations and faithful facial details.

## 4.5 Parameters and Inference Time Comparison

The parameters size and inference time are critical to the deployment of the model in real-world scenarios. Here, we compared the parameters size and inference time of different methods on an NVIDIA A100 GPU. To ensure fairness, we set the batch size of each method to 1 and conducted five evaluations for average. The corresponding statistics of different methods are reported in Table 4. As can be observed, our method significantly outperforms the other methods in both model size and inference time, demonstrating the effectiveness and efficiency of our approach. This also confirms the potential of our method for practical real-time applications.

## 4.6 Limitations and Future Work

Although our framework offers an effective solution to portrait shadow removal. It may face challenges when processing dark-skinned or non-uniform illumination faces. For the former, the network may confuse skin color with shadows. For the latter, the network may exploit the lighting information from local regions (*e.g.*, regions with specular illuminations) to assist in rectification of global illumination, leading to color shift (see Fig. 9). Addressing this limitation, possibly through training with more dark-skinned

**Table 4: Parameter size and inference time comparisons among different methods.**

| Method | Params. (M) ↓ | Infer. Time (ms) ↓ |
|---|---|---|
| He *et al.* [12] | 61.11 | 320000.0 |
| Cun *et al.* [5] | 137.18 | 90.6 |
| Hu *et al.* [15] | 122.49 | 48.8 |
| Liu *et al.* [29] | 10.25 | 2310.0 |
| DMTN [27] | 27.90 | 84.4 |
| TBRNet [28] | 46.71 | 83.9 |
| Ours | **3.37** | **28.0** |

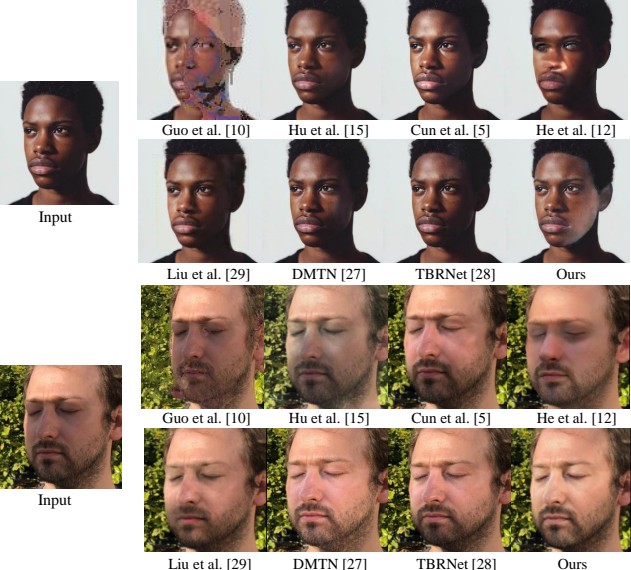

**Figure 9: Failure cases of our method with color shift when processing dark-skinned or non-uniform illumination faces.**

samples or involving external uniform illumination references, is a goal for future work.

## 5 Conclusions

In our paper, we present the Self-Exemplar Illumination Equalization Network for portrait shadow removal. Utilizing Self-Exemplar Illumination Equalization Blocks, our network rectifies shadow illumination through self-exemplar feature matching. This process generates illumination-corrected features and spatial matrices, enhancing the equalization of facial feature illumination. Our method demonstrates superior performance in shadow removal and facial texture recovery on two datasets.

## Acknowledgments

The work is supported by Guangdong Basic and Applied Basic Research Foundation (2024A1515011995), the Guangdong Natural Science Funds for Distinguished Young Scholar under Grant 2023B1515020097, and National Research Foundation Singapore under the AI Singapore Programme under Grant AISG3-GV-2023-011.

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
