# OpenReview forum: "Portrait Shadow Removal via Self-Exemplar Illumination Equalization"
_acmmm.org/ACMMM/2024/Conference — MM2024 Oral_

### Official Review · Reviewer_iXd7 · 2024-05-17

**Rating:** 5
**Confidence:** 3

**Summary:**

This research proposes a novel progressive facial shadow removal network (SExmNet) that leverages internal illumination exemplars. The paper is well-written and easy to understand, with elegant technical solutions and substantial experiments. However, some details need improvement.

**Strengths:**

Pros:
1. The paper is well-motivated and focused on utilizing internal illumination exemplars to generate realistic, shadow-free facial images.
2. The paper proposes an elegant method with extensive experimental validation.

**Limitations:**

Cons:
1. Figures 6 and 7 strangely exclude ground-truth examples.
2. This paper does not discuss the method's limitations, which keeps showcasing its advantages in Section 4.6.
3. In Section 3.2, the rationale for using rematched features to modulate original features needs further clarification. In my opinion, this modulation mechanism usually involves external priors rather than intermediate features.
4. In the related works of illumination compensation, the reasons for the proposed method's advantage are unclear.
5. In the related works of matching correspondence, the claimed causal relationship between the unsupervised matching correspondence and the improved removal quality is difficult to find convincing.
6. The related work section's structure is confusing, with three subheadings not paralleled with each other.

**Suitability:**

2

---

### Official Review · Reviewer_yySa · 2024-05-24

**Rating:** 4
**Confidence:** 4

**Summary:**

This paper presents a new Self-Exemplar Illumination Equalization Network that proposes using non-shadowed areas as lighting references to balance shadowed areas, and achieves facial shadow removal using self-exemplar feature matching module and feature-level illumination rectification module.

**Strengths:**

The paper clearly outlines the problem it aims to solve and provides detailed descriptions of the proposed model, which helps to understand the method proposed in the paper. Additionally, the experimental results are excellent, and the video in the supplementary materials provides a clear demonstration of the effectiveness of real-time facial shadow removal.

**Limitations:**

(1)	Regarding the Correspondence Estimator operation in Self-exemplar Feature Matching Module, although its effectiveness is evident from Figure 3, the implementation steps are unclear. Could you please elaborate on the implementation steps?
(2)	This work provides a novel approach to preserving facial details while eliminating facial shadows. However, the method relies on non-shadow regions as references. If the entire facial area is in shadow, would this method still be effective?
(3)	As stated in the paper, the diverse shadow conditions and facial skin tones in portrait images make it difficult for these methods to learn accurate and consistent mappings. Therefore, more details of the experimental dataset should be disclosed, including the proportions of different ethnicities and scales.
(4)	I believe that the visualization experiments should also include the removal of shadows from faces of diversified colored individuals, to highlight the model's generalization capability in the task of facial shadow removal.
Finally, I believe that the task of facial shadow removal overlaps with the task of low-light facial image enhancement. Have the authors considered selecting models from the low-light image enhancement domain for comparison, in order to highlight the superiority of the proposed method?

**Suitability:**

2

---

### Official Review · Reviewer_YRk7 · 2024-06-02

**Rating:** 3
**Confidence:** 3

**Summary:**

This paper proposes a  Self-Exemplar Illumination Equalization Network for the task of Portrait Shadow Removal.
It is claimed firstly use  internal feature matching for portrait shadow removal.

**Strengths:**

The testing results of videos show that the proposed method has a relative stable result.

**Limitations:**

1、The authors claimed that there is no publicly available large-scale paired
shadow/shadow-free dataset.  However, the related database can be accessed from ref[29].
Why don't use it for comparison,including training and testing?
2、The authors created a dataset with 3,000 pairs of shadow/shadow-free portrait images for training.
Are all the compared methods trained on this dataset?
3、Some results of compared methods from ref[29] are the same with those of ref[48].
 However,  the results provided in this manuscript  are not the same with the results of ref [12],[15],[5],[29].
The SSIM on UCB dataset  in ref[29] is 0.866, but the result in this manuscript is 0.851.
 The authors should explain why?

**Suitability:**

2

---

### Official Review · Reviewer_fqQt · 2024-06-02

**Rating:** 5
**Confidence:** 3

**Summary:**

The paper introduces a Self-Exemplar Illumination Equalization Network for portrait shadow removal.
Based on the observation that partially shadowed portraits can find ideal exemplars within their non-shadowed facial regions, the proposed approach utilizes non-shadowed regions as an illumination indicator to equalize the shadowed regions, generating deshadowed results without boundary-merging artifacts.
The method comprises cascaded Self-Exemplar Illumination Equalization Blocks, each containing a self-exemplar feature matching module and a feature-level illumination rectification module, which are responsible for explicitly leveraging the internal illumination exemplars for relighting the shadow region, and rectifying the illumination of the input facial features, respectively.

**Strengths:**

The experimental results is amazing. And the model is simple but efficient.

**Limitations:**

Some important details of the model are missing:
(1) What is the feature extractor?
(2) What is the correspondence estimator?

**Suitability:**

3

---

### Meta-Review · Area_Chair_jeTD · 2024-07-01

**Recommendation:** Accept (Oral)
**Confidence:** 5

**Metareview:**

This paper presents an approach to portrait shadow removal by utilizing self-exemplar within non-shadowed regions. Reviewers find the presented approach is novel, and the experimental results are outstanding. The rebuttal has addressed the concerns of the reviewers, and all the reviewers give Weak Accept as the rebuttal addressed their concerns. Thus, the AC recommends acceptance of the work.